# A Recommender System for Robust Smart Contract Template Classification

**DOI:** 10.3390/s23020639

**Published:** 2023-01-05

**Authors:** Sandi Gec, Vlado Stankovski, Dejan Lavbič, Petar Kochovski

**Affiliations:** Faculty of Computer and Information Science, University of Ljubljana, 1000 Ljubljana, Slovenia

**Keywords:** smart contract, classification, cluster, recommender system, inheritance

## Abstract

IoT environments are becoming increasingly heterogeneous in terms of their distributions and included entities by collaboratively involving not only data centers known from Cloud computing but also the different types of third-party entities that can provide computing resources. To transparently provide such resources and facilitate trust between the involved entities, it is necessary to develop and implement smart contracts. However, when developing smart contracts, developers face many challenges and concerns, such as security, contracts’ correctness, a lack of documentation and/or design patterns, and others. To address this problem, we propose a new recommender system to facilitate the development and implementation of low-cost EVM-enabled smart contracts. The recommender system’s algorithm provides the smart contract developer with smart contract templates that match their requirements and that are relevant to the typology of the fog architecture. It mainly relies on OpenZeppelin, a modular, reusable, and secure smart contract library that we use when classifying the smart contracts. The evaluation results indicate that by using our solution, the smart contracts’ development times are overall reduced. Moreover, such smart contracts are sustainable for fog-computing IoT environments and applications in low-cost EVM-based ledgers. The recommender system has been successfully implemented in the ONTOCHAIN ecosystem, thus presenting its applicability.

## 1. Introduction

The Internet of Things (IoT) is a modern technology that allows various devices to exchange information over the Internet. These devices aid their users in facilitating their everyday routines and automate them via digitalization. Typical IoT environments comprise many components, such as sensors, gateways, computing, and storage nodes in the Edge-to-Cloud continuum. Despite the enormous benefits the IoT has introduced to domains, such as smart and safe construction, Industry 4.0, smart cities, and IoT solutions are generally based on centralized architectures. Essentially, a single point of failure in such scenarios, where the server goes down, can significantly reduce the Quality of Service (QoS) and the Quality of Experience (QoE), decrease trust in the system, and place its IoT data in danger. However, these problems in IoT can be overcome by implementing blockchain and its core technologies (i.e., immutable ledgers, smart contracts, and smart oracles) [1]. Thus, the usage of smart contracts in IoT environments not only benefits from the implicit blockchain benefits but enables the implementation of comprehensive functionalities that may significantly increase the distribution and overall trust of the IoT environment.

A smart contract is a digital variation of a traditional contract stored as programs on a blockchain that is ordinarily used to automate the execution of an agreement without any intermediary’s involvement. The basic purpose of each smart contract is to automate workflows by triggering specific actions when conditions are met. Our work relies on the first publicly released smart contract, enabling ledger Ethereum with built-in fully fledged Turing-complete programming language Solidity. Nevertheless, the operational cost and speed limitations in block generation times using native Ethereum ledgers cannot always satisfy the requirements of many use case domains, including IoT. For this reason, new solutions using the Ethereum ledger core computational engine called the Ethereum Virtual Machine (EVM) launched their own ledger. The primary purpose of these ledgers is to increase sustainability and overall performances. Since general blockchain research studies strive to improve general requirements such as the optimization of query processing [2,3] and many others, the smart contract is more likely to investigate how to cover new use cases from the cloud to the edge using the development of more performant blockchain ledgers [4]. Moreover, on-chain operations started interacting with off-chain data within the Smart Oracle mechanism available in smart contracts.

The development process of smart contracts in Solidity language is a very challenging task. For example, the documentation is limited, the community of developers is limited, and the language does not follow standardized design patterns, etc. Since the principles of smart contracts do not allow updating or deleting a smart contract once deployed, the functionalities have to be extremely well written and tested to ensure security, as outlined by Zou et al. [5].

The high-level goal of our work is to decentralize fog-based IoT architectures with the integration of smart contracts on the fog layer using low-cost EVM-enabling ledgers. To reach the goal, we propose a recommender system that provides the developer with robust, reusable smart contract templates based on the fog architecture characteristics and requirements. Our approach can be summarized as follows. First, we obtain production-ready and secure smart contract templates (e.g., OpenZeppelin (https://www.openzeppelin.com/ (accessed on 3 November 2022)) and ChainLink https://chain.link/ (accessed on 14 October 2022)) and classify them according to the interactions that occur among them and the actual purpose of the contracts. Afterward, we define our fog architecture in a standardized TOSCA format [6]. Then, we use our TOSCA-defined fog architecture and requirements as the input data of our recommender system. As a result, we obtained a list of the relevant smart contract templates that can be used in our system directly or in an extended manner to fulfill all use-case-specific requirements and workflow. By providing such templates, we implicitly promote secure smart contracts by design and strive to follow good practices provided as ready-to-use functions or interface function standards. Hence, the development time of the developers is reduced. By upgrading fog architecture with smart contracts, it is possible, for example, to provide a pricing policy among end users that want to provide the hardware and/or software resources within the fog environment using cryptocurrencies or tokens. A main benefit for the system is either a reduction in the operational cost and/or the increase in the system’s revenue. To summarize, we provide the following contributions in the paper:We propose a classification of smart contracts built upon the characteristics of the contracts and the inheritance relationship among them.We develop a recommender system algorithm for smart contract template selection in fog architectures.We introduce a novel architecture for a recommender system that proposes robust smart contract templates based on the fog architecture and system requirements’ input data.

The remainder of the paper is split into seven sections. Section 2 places our work within the context of other related works. Section 3 describes the baseline scenario. Section 4 analyzes reusable smart contract templates by using classification and clustering methods to propose fog-oriented classification. Section 5 proposes the recommender system of the classified smart contract templates for fog architectures. Section 6 presents an experimental evaluation of the proposed recommender system, and Section 7 draws the conclusion.

## 2. Related Works

Smart contracts were first publicly available in 2015 with the release of the Ethereum decentralized ledger (https://ethereum.org (accessed on 2 October 2022)). The novel concept enabled application scripts in terms of their deployment and execution in a safe environment named the Ethereum Virtual Machine (EVM) on distributed nodes in the form of decentralized applications (DApps). In the Cloud domain, smart contracts became extremely popular as a tool to improve existing environments [7,8,9] and even addressed different requirements such as privacy-preserving data sharing [10]. With the introduction of dedicated optimized variations of Ethereum (e.g., BNB chain (https://www.bnbchain.org (accessed on 3 October 2022)), Polygon (https://polygon.technology/ (accessed on 7 October 2022)), Optimism (https://www.optimism.io/ (accessed on 9 October 2022)), and many others) using the same EVM core engine, the sustainability of the smart contract use case increased. Even IoT-dedicated ledgers such as IOTA (https://www.iota.org/ (accessed on 16 October 2022)) started with the integration of EVM, enabling ledgers with limited compatibility and thus allowing the integration of smart contracts into more complex IoT scenarios [11]. Lakhan et al. [12] proposed an Ethereum smart-contract-based client-fog-cloud healthcare system by integrating sustainable smart contracts for dedicated operations focused on scheduling.

The development of smart contracts enabling Cloud-to-Edge architectures is a complex task that requires experienced developers in combination with the usage of analysis tools [13] to avoid or minimize the known vulnerabilities in the design process of smart contracts. To better understand the design of smart contracts in the Ethereum ledger, Angelo et al. [14] performed a study of the smart contracts’ similarities by analyzing the design of their interfaces and grouping them into a small set of clusters. The relevance of code reutilization was further studied by Chen et al. [15], and the authors found that 26% of the contract code blocks were reused until early 2021. In collaboration with the EVM-based ledger solutions, individual researchers and companies, such as OpenZeppelin, actively started to promote the usage of production-ready smart contract templates in the development process. Moreover, a particular focus on understanding tokenomics was proposed as a design-oriented morphological framework by Freni et al. [16].

A recommender system is a subclass of information systems. In general, the workflow of recommender systems can be described with three steps: (i) information collection phase, (ii) learning (algorithmic) phase, and (iii) recommendation phase [17]. In the context of the blockchain domain, there were many variations in the implementation of recommender systems in a distributed manner [18]. Zhang et al. [19] developed a recommender system tool CloudRecommender by proposing a declarative solution for recommending Cloud infrastructure services in contrast to our approach of recommending smart contract templates. A context-free recommender system approach was proposed by Lisi et al. [20], and the authors offloaded most of the system’s logic into smart contracts, thus making it mainly decentralized. On the other hand, the scope of our recommender system is domain-specific, and smart contracts have played the role of recommender output results in order to improve IoT fog architectures with blockchain-implicit properties and/or dedicated functionalities.

All code-based analysis studies and available smart contract templates (e.g., OpenZeppelin and our past work) comprised the foundations of the further classification of the smart contract templates proposed in our work. To our knowledge, no studies on recommender systems for robust smart contract templates in the context of fog computing and the Internet of Things (IoT) have been proposed.

## 3. Baseline Scenario

Typical IoT applications are composed of many different components such as sensors, gateways, computing, and storage nodes that are often not owned by the application developer but implemented as pay-per-use services by third-party resource providers. Due to the immense diversity of IoT environments, they often differ because of their specific requirements. Moreover, applying blockchain as a technology in such complex environments can cause lengthy development, reduce scalability, and increase complexity due to incompatible standards.

The baseline scenario (see Figure 1) of our solution is motivated by the above-elaborated problems and envisions the implementation and utilization of a novel blockchain recommender system that will perform the following: (1) compose smart contract templates that are based on the application’s specification, following good practices and design patterns; (2) facilitate the blockchain implementation by providing the IoT developers with reliable, standardized smart contracts for their use cases; (3) accelerate the overall development and implementation of smart contracts in IoT environments. Moreover, it will enable peer-to-peer (P2P) transactions between the application’s developers and the resource providers. In other words, the scenario currently targets two essential groups of potential users: The IoT application developers and the resource providers. The complete workflow is composed of five consecutive steps:1.The application developer defines IoT’s available application quality requirements (e.g., data access, accessibility, scalability, etc.) and baseline cloud–fog specifications for application deployment using a TOSCA standard.2.The Recommender System receives the requirements and specifications as input and assembles a set of smart contract templates, which will be further selected by the application developer and further implemented in comprehensive fully functional dedicated smart contracts.3.The Application Developer is offered the opportunity to approve or modify the smart contracts. Once the smart contracts are approved, they are deployed on the blockchain.4.When the smart contracts are deployed on the blockchain, different entities in the system (e.g., resource providers) can interact with them according to the democratic voting that takes places among the system’s stakeholders and resource providers. For instance, by interacting with the smart contract, the resource providers enable access to their resources (i.e., computing, storage resources, and IoT data) for third parties and obtain direct incentives from that interaction.5.When the interaction with the smart contracts results in new records on the blockchain, the application’s developer (i.e., the IoT application) is allowed to interact with the providers’ resources based on the policy agreed in the previous step.

## 4. High-Level Classification of Reusable Smart Contracts

This section presents a classification of reusable smart contracts based on hierarchical analysis. First, we offer the main domains in the ONTOCHAIN environment. Then, we select the most currently advanced library for reusable smart contracts and describe them by their base properties and predefined modules. By the topology of smart contracts and inheritance among the smart contracts and in contrast to ONTOCHAIN domains, we propose a classification of smart contracts. Since tokenomics represents one of the most important categories, we present the token standards and summarize the fundamental properties.

### 4.1. Pillar Domains in the ONTOCHAIN Ecosystem

The next-generation ONTOCHAIN framework is a modular blockchain framework that leads to a more human-centered Internet that supports the values of openness, decentralization, inclusiveness, and the protection of privacy [21]. It delivers various real-world solutions, such as trustworthy web and social media, trustworthy crowdsensing and reputation management, distribution logistics, data management, and similar solutions via the use of multiple ledgers and semantic technologies. Its main goal is to further expand as an ecosystem that will address and complement different use cases using the ONTOCHAIN protocols and their different blockchain and semantic components.

Figure 2 depicts the high-level multi-layer architecture of the ONTOCHAIN framework. The blockchain layer provides a distributed execution environment that enables access to multiple distributed ledger networks that the ecosystem stakeholders can utilize. The ontologies layer comprises novel ontologies that can be managed in a trusted and secure manner. The application and core protocols and services enable seamless interaction between the framework’s layers. Namely, these interoperability protocols offer identity management, reputation management, data provenance, market mechanisms, and other functionalities that the applications of the ecosystem can exploit.

Based on meticulous state-of-the-art analyses [22,23], the ONTOCHAIN framework identified 15 domains that can deliver a plethora of core use-case applications that can complement the ecosystem and exploit the framework’s functionalities. Namely, the following application domains were identified: agrifood, art, construction industry, education and science, energy management, fashion and luxury, financing, healthcare, Industry 4.0, information fact-checking, insurance, logistics mobility, public administration, and tourism. Given the uniqueness and variety of different types of possible applications, their decentralization and development of smart contracts can be complex and lengthy. Motivated by this, the following sections introduce a categorization of smart contracts and a recommender system for smart contracts that aims to facilitate smart contract development and application decentralization.

### 4.2. Base Categorization of Smart Contracts

Initially, smart contracts were represented in a low-level, assembly-like sequence of operands that demanded proper low-level programming knowledge and experience from developers. In March 2018, the first version of Solidity-based, JavaScript-like programming language, smart contracts, was released and thus encouraged a more comprehensive range of developers to contribute to the Ethereum ecosystem. The development methodology did not fully follow standard processes due to the specifics in the life-cycle of Ethereum smart contracts [24]. For example, it is impossible to update an instance of a smart contract by only redeploying it due to the tamper-proof implicit properties of the Ethereum ledger. Therefore, coding anomalies in the smart contracts, such as bugs, lack of validation, incompatible command sequence, and other issues, led to security vulnerabilities with usually severe consequences in communities using vulnerable smart contracts [25]. This iterative learning process delivered coding guidance provided by various analysis tools [13] and design patterns in the form of smart contract templates. The latter covers different functionalities, standards (e.g., tokens), and good practices that may facilitate the development process of solidity-based smart contracts. Most companies providing production-ready smart contracts rely on security-audited smart contract templates by OpenZeppelin.

The library of modular, reusable, and secure smart contracts for Ethereum and other EVM-enabling ledgers consists, in the time of writing (library version 4.8.0), of 174 smart contracts. The actual number of operational ones is 155, defined by the base smart-contract document definitions such as contract, abstract contract, interface, or library. By default, the smart contracts are organized into 11 main modules that may be summarized as follows:The Access module supports basic, role-based, cross-chain, and other access control mechanism or policies: typically, these policies are applied in the smart contract’s individual functions.The Cross Chain module provides a component to improve cross-chain awareness of smart contracts via the Arbitrary Message Bridge (AMB) mechanism.The Finance module roughly includes functionalities for financial systems on both types of assets, ETH or other EVM coins and tokens. The main two contracts provide splitting payments among multiple entities and the vesting of assets for a given beneficiary.The Governance module consists of comprehensive on-chain governance smart contracts for use cases such as voting, governance control, and other time-lock-related functionalities.The Interfaces module summarized the interfaces that may be implemented in smart contracts and thus provided dedicated functionality workflows.The Meta-transactions module includes minimal-extended ERC-2771 contract instance and a context variant with the ERC-2771 support to support gasless transactions.The Proxy module defining a low-level set of contracts defining different proxy patterns: (i) not able to upgrade, immutable by default, and (ii) scalable using an upgradable proxy pattern.The Security module aims to cover the security domain—More concretely, the good practices enhancing security.The Tokens module includes the definition of fundamental ERC-based token standards that are presented and compared among them in the following subsection.The Utilities module consists of miscellaneous contracts and libraries containing utility functions that facilitate the data management of new data types, security, and the safe use of low-level primitives.The Vendor module includes smart contracts that enable work with the most common EVM-compliant chains (e.g., Arbitrum, Polygon, Optimism, etc.).

The main categorization based on the described modules above consists of smart contracts proposed by the library authors consisting of different types of smart contracts and smart contract quantitative coverage, as depicted in Figure 3. Our classification is derived from the main categories relying on inheritance levels and interconnections from the perspective of available functionalities, as presented in the following subsections.

### 4.3. Tokenomics

Tokenomics plays a crucial part in the design of smart contract functionalities due to the possibility of providing advanced functionalities enabled by the token exchange and/or interaction throughout smart contracts. EVM-enabling ledger tokens, by design, run on a Layer-2 protocol that relies on Layer-1 for security and consensus. For example, all operations performed with tokens are fueled in the transactions by native cryptocurrencies, such as ETH in the Ethereum ledger. In the smart contract implementation of token functionalities, it is vital to understand the token standards, their methodology, capabilities, and limitations. Currently, there are five token standards officially approved by the Ethereum developer community in addition to the attempts that were not approved or finalized, such as ERC-223 and ERC-1337. The main comparison of the official ERC standards is summarized in Table 1.

The standards are mainly divided by fungible properties along with other specific properties. In our work, we consider token standards that contrast with common use cases that may be applied to fog architectures.

### 4.4. Hierarchical Analysis of Smart Contracts

The design of smart contracts follows standard object-oriented programming features, such as an extension of the functionalities of a program to encourage the development of individual ready-to-use modules in a comprehensive smart contract. For example, developers strive to more systematically separate codes, reduce the dependencies, import directives, increase the re-usability of existing programming code, and even enforce the proposed workflow (e.g., voting smart contract). In the initial OpenZeppelin analysis, 65 individual smart contracts and 90 smart contracts were involved in the inheritance. The inheritance is represented either as extended in a smart contract or defines the module’s fundamental behavior, functions, or primitives, identifying four main clusters sorted by the ascending number of contracts in the interaction:1.The first cluster consists of two smart contracts designed for EVM-based dedicated chain Arbitrum. The hierarchy dictates the interface contract (*IInbox*) extending standard Arbitrum events through the base interface (*IDelayedMessageProvider*) as shown in Figure 4 in the bottom left.2.The second cluster organizes the ERC-1820 standard defined in the contract*ERC1820Implementer*, extending the interface *IERC1820Implementer* and roughly defining a universal registry smart contract where the address policy is defined (e.g., an address can register which interface it supports and which smart contract is responsible for its implementation), as depicted in Figure 4 in the bottom upper left.3.The third cluster of seven contracts describes the proxy policy for different purposes (see Figure 4: bottom right).4.The fourth biggest cluster of ninety-nine contracts involves ERC token standards in conjunction with the modules such as governance, access, and others, as shown in Figure 4. The most dominant module in inheritance is token, followed by governance and utils (utilities). Deployable contracts are realized by inherited abstract contracts containing minimal business logic and interfaces that enforce function definitions, including global variables and events.

We classified the most relevant smart contract functionalities from the basic OpenZeppelin categorization and clustering derived from the inheritance among smart contracts. In addition, the ONTOCHAIN environment was considered in order to converge the categories into requirement-like descriptions on two levels (see Figure 5): (i) the base classifier and (ii) the detailed classifiers suitable in the process of the recommender system where the system requirements are more concrete.

## 5. Recommender System of Reusable Smart Contracts for Fog Architectures

In this section, we describe the recommender system’s methodology. First, we consider the fog environment’s data representation to represent the architecture’s base properties, such as components, end users, and relationships among them. Then, we propose a taxonomy of the input data described as fog architectures and related requirements defined by the stakeholders of the environment. At the end of the section, we propose the architecture of our recommender system.

### 5.1. Data Descriptors

Fog architectures consist of many types of components that are represented in different formats and even different notations. In the cloud computing domain, this problem was addressed in 2014 with the establishment of a standard proposed by a nonprofit consortium OASIS (www.oasis-open.org (accessed on 28 September 2022)) named Topology and Orchestration Specification for Cloud Applications (TOSCA). The standard specifications defined the cloud architectures with components in YAML format. Furthermore, the research community also addressed other emerging domains, such as fog and edge, where Tsagkaropoulos et al. [26] proposed extensions to the existing standard. In our work, the fog architectures are packed in TOSCA format to describe the interactions among the components and the involved entities (e.g., properties, capabilities, and relationship), as shown in Listing 1.

**Listing****1.** Example of a component with basic properties in TOSCA.


                tosca_definitions_version: alien_dsl_2_0_0
		topology_template:
		node_templates:
                node_templates:
		Componnet_A:
		type: org.alien4cloud.fog.nodes.CustomFogNode
		properties:
		duration: 55
		variation: 20
		log_length: 2000
		requirements:
		- endpoint:
		node: Component_B
		capability: org.myFog.capabilities.FogComponentEndpoint
		relationship: org.myFog.relationships.FogC_ConnectToComponent


### 5.2. Architecture of the Recommender System

Fog architecture is becoming popular due to its possibility for offloading computational tasks, and thus, the components on the fog nodes that can be a part of cloud-dedicated data centers and infrastructure, provided by the end users [27]. The process of offloading includes end-user provisioning services or resources in the fog environment and ordinarily demanded manual operations (e.g., agreement about the price and policy, registering a new provisioning component, etc.) that are often performed via centralized components. These manual operations can be defined in a dedicated smart contract following the requirements of the system and stakeholders. Developing such complex smart contracts is very difficult and requires an experienced smart contract developer. Therefore, the development can be facilitated since the recommender system provides, as a result, smart contract templates that are intended to be extended to follow the use case of the system jointly with smart contracts.

The architectural design of the recommender system, depicted in Figure 6, relies on two main inputs: (i) fog architecture represented in standardized format *TOSCA v. 2.0* and (ii) requirements provided as system requirements by the fog architect. The recommender system engine analyzes the input data using the three following main steps:1.The architecture reasoning step analyzes the topology of the fog architecture among the individual sets of components by focusing on the capability attribute, which defines the characteristics of the specific component, and the relationship that provides the type of the connection (e.g., protocol, allowance such as command-line interface (CLI), etc.). Additional features are extracted from the fog environment’s high-level properties, such as the types and number of components where the component’s representation and the algorithm’s details are presented in the following section.2.The requirement analysis step focuses on system requirements that can be either general or detailed. General requirements such as token support provide a wider range of smart contract template candidates, which include all types of tokens (fungible and non-fungible). The detailed requirements, on the other side, significantly narrow the smart contract template candidates.3.The comprehensive check step unifies the results from the first two steps, particularly the cases of conflicting smart contract templates (e.g., ER-C20 and ERC-721) that cannot be defined in the same smart contract together. Therefore, the first and second steps are repeated, including scoring on each smart contract template candidate, where the one with a higher score is included in the component’s result list.

The algorithm of the recommender system engine is presented in the following section. Finally, the developer obtains a list of smart contract templates by matching the fog architecture’s typology and system requirements. The developer obtains ready-to-use smart contracts, abstract smart contracts that need to be extended, interfaces, and libraries implicitly via other contracts.

## 6. Experimental Evaluation

### 6.1. Cost Estimation of Smart Contracts

The long-term sustainability of smart contracts integrated with the fog environments is a significant property in developing dedicated smart contracts based on robust templates. The aim of this experimental study is cost estimation expressed with Ethereum network fees *Gas* by definition, which measure the amount of computational effort required to perform specific operations on Ethereum or other EVM-compliant ledgers. In our work, we use Ethereum’s Gas represented with the unit *Gwei* or in *nanoeth*, which is represented as $1ETH=10^{9}Gwei$.

Our cost estimation methodology follows the following steps: (i) deployment of smart contract in a local testnet environment, (ii) the execution of all available smart contract functions with dedicated gas estimation function or empirically execute complex functions [28], and a (iii) summary of the operational cost obtained in the first and second steps. We use the default *Gas* parameters within the TruffleSuite (https://trufflesuite.com/ (accessed on 14 October 2022)) framework to simulate the EVM environment. The experiments were performed on deployable smart contracts of type contract and the simulation’s results are shown in Figure 7. The aim of the experiment is to prove the sustainability of smart contracts within the IoT fog environment.

### 6.2. Fog Recommender System Algorithm

The recommender system’s algorithmic basis on fog architectures that are used as baseline inputs are obtained by the largest Cloud providers: Azure (https://docs.microsoft.com/en-us/azure/architecture/reference-architectures/ (accessed on 4 September 2022)) provides categorized cloud architecture references in our IoT case and Google Cloud (https://cloud.google.com/architecture (accessed on 4 September 2022)) in the cloud architecture center offers reference to IoT architectures with guidance, and Amazon AWS (https://aws.amazon.com/architecture/well-architected/ (accessed on 6 September 2022)) provides defined cloud-to-edge architecture applications. We composed our fog architectures from the cloud providers and available architectures that will be run on our Algorithm 1 with different requirements. The algorithm is written in JavaScript programming language as REST API services where the time complexity of the algorithm is quadratic, $O(n^{2})$. In the worst-case scenario, the algorithm based on two nested loops has to perform *n* operations per iteration of the outer loop, making it a total of n*n operations. Since the order of magnitude is up to 100 components per IoT fog architecture, this does not represent an issue. The base data (available components, classifications, and smart contract templates) are stored in the MongoDB database.
**Algorithm****1** Recommender system algorithm represented in pseudocode**procedure** Recommender system     templates←dictionaryofsmartcontracttemplates     fogTOSCA←architecturerepresentedinTOSCAformat     requirements←listofrequirements     resultList←resultlistofsmartcontracttemplates     finished←false     scoring←false    **while**
finished=false||scoring=true
**do**    ▹ while the results are not conflicting         **if**
scoring==true
**then**  ▹ Loop through the *fogTOSCA* components and vote (relationship, capability, type)              **if** scoring results not equal **then**                   **return** resultList and label conflicting templates              **end if**              **if** scoring results are equal **then**                   **return** resultList and label conflicting templates              **end if**        **end if**       **while** iterate through *templates* **do** ▹ filter each contract based on the requirements                  ▹ if conflicting contract found set *scoring* = true         **end while**       **while** iterate through *fogTOSCA* **do**     ▹ for each component, add weight scores about relationship, capability and component type to potential templates                     ▹ if conflicts still occur set *scoring* = true          **end while****     end while****end procedure**

Moreover, to compose the fog architectures used in our experimental study, we defined each architecture in TOSCA v. 2.0 format by using a Web tool Alien4Cloud (https://alien4cloud.github.io/index.html (accessed on 8 September 2022)), which provides a series of more than 90 representations of the standard definitions of cloud-to-edge components such as hypervisor, deployment, network, server, storage, management, service, gateway, and others. As outlined in the previous section for the main steps of the architecture, the algorithm extracts the key characteristics of the architecture, such as relationship, capability, and type. The algorithm mainly relies on the component’s type and relationship related to the three capability properties: feature, host, and scalability. These features are aligned with identified classifiers. Since there are no research baseline architectures to be used, we evaluated our recommender system’s algorithm with simulations.

We performed simulations of our recommender algorithm on three base architectures (Azure IoT reference architecture (https://tinyurl.com/mrp8hjdd (accessed on 12 September 2022)), Intelligent Products Essentials reference architecture (https://tinyurl.com/3m9vnpba (accessed on 14 September 2022)), and Smart Metering for Water Utilities (https://tinyurl.com/4n4ny2ea (accessed on 17 September 2022)) using different requirements, as shown in Table 2. We did not include inherited contracts in the smart contract’s template results because they are implicitly included.

### 6.3. Smart Contract Development Time Evaluation

To evaluate the performance of the recommender system in the context of the smart contracts’ development time, in December 2022, we prepared a study with 50 participants with different levels of blockchain expertise. We targeted participants to whom we had direct access, such as undergraduate university students, researchers, and collaborators in the ONTOCHAIN project. Each participant had to develop four smart contracts, and each participant had to develop the smart contracts once without the help of the recommender system and also by using the recommender system. To maintain the moment of surprise, the order in which they received the assignments was random.

At the time of the result’s analysis, we excluded the participants who did not submit all four smart contracts, did not pass the unit tests for the developed smart contracts, or took more than two hours for development. As a result, we finalized the evaluation with the 36 participants passing the tasks, 27.8% female and 72.2% male, with ages ranging from 20 to 55 years old. In general, the participants were 36.1% undergraduate students and 63.9% non-students.

The results have shown that for the smart contracts that were developed without the help of the recommender system, the participants spent an average of 50 min on development. In comparison, for the smart contracts that were developed using the recommender system, they developed smart contracts with an average of 12 min, which reduced the development time by 76%.

## 7. Discussion and Conclusions

In our work, we proposed a recommender system that will simplify and accelerate the development of decentralized IoT applications. The recommender system outputs smart contract templates based on the system’s requirements that the IoT developer defines during the development phase. In our experimental results, we focused on two key properties: sustainability and development time reduction.

Smart contracts initially interact with the EVM-based ledger first in the development process, which is the most costly operation. This is expected since the entire Dapp as a programming script has to be deployed on the ledger. Once deployed, the involved stakeholders of the system trigger functions that update the state of the variables, execute transactions, or other purposes. Functions are compared to the deployment of the contract, and they are significantly cheaper. Using a low-cost EVM-enabling ledger such as Polygon makes it possible to perform most functions for less than USD 0.10 per transaction. Thus, enabling smart contracts in fog environments is sustainable.

The quantitative results of our recommender system were addressed in the analysis of the results provided by the algorithm. The workflow of the algorithm considers the topology of the fog architecture with specified requirements. Based on our simulation of three heterogeneous IoT reference architectures, the system provided a set of relevant smart contract templates. These templates can be merged, extended, or modified to comply with the environment. The development process is simplified, and the overall smart contract preparation time is reduced.

Although this work addresses important challenges for building decentralized IoT applications by using smart contract templates proposed by the newly developed smart contract recommender system, there are still many challenges to be tackled in the future. For instance, the recommendation results may be significantly improved if a reputation system is integrated into the recommender system. As such, the recommender system could also categorize the smart contracts and choose standards based on their reputation that can be based on various requirements (e.g., security, data safety, scalability, etc.). Another important challenge in decentralizing IoT applications would be the management of decisions and service-level agreements in larger IoT systems, where various service providers play important roles in the provision of sensor data and in computing and processing resources. By expanding the recommender system functionalities, to update the smart contract templates based on democratic methods for reaching a consensus, the recommender system will evolve into a decentralized autonomous organization. This will allow seamless smart contract management operations in complex IoT scenarios, where all relevant stakeholders would be allowed to participate in smart contract management without having prior expertise in smart contract development. These challenges will be further researched within the scope of the ONTOCHAIN project, and the results will be presented in our future works. 

## Figures and Tables

**Figure 1 sensors-23-00639-f001:**
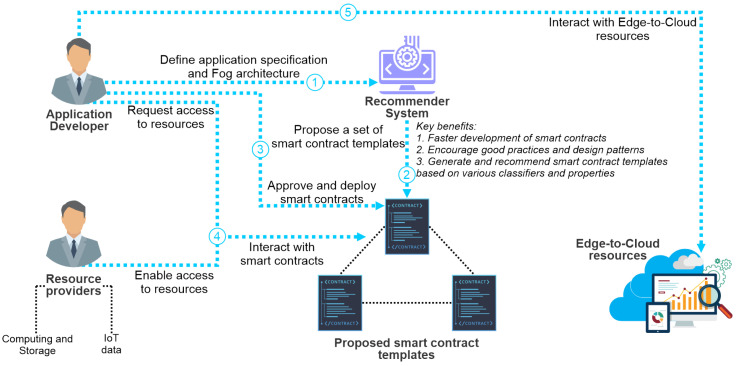
Outline of the baseline scenario.

**Figure 2 sensors-23-00639-f002:**
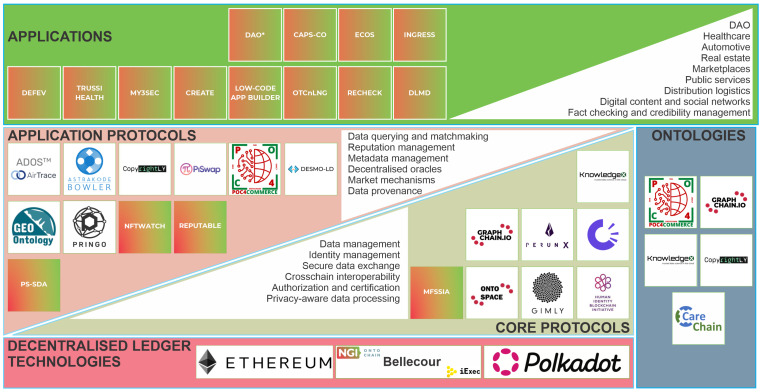
ONTOCHAIN high-level architecture.

**Figure 3 sensors-23-00639-f003:**
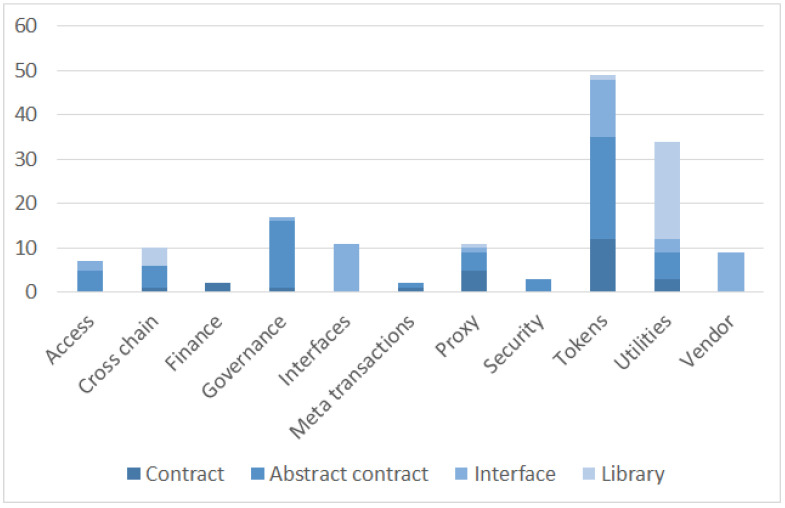
OpenZeppelin main smart contract modules in relation to the contract types.

**Figure 4 sensors-23-00639-f004:**
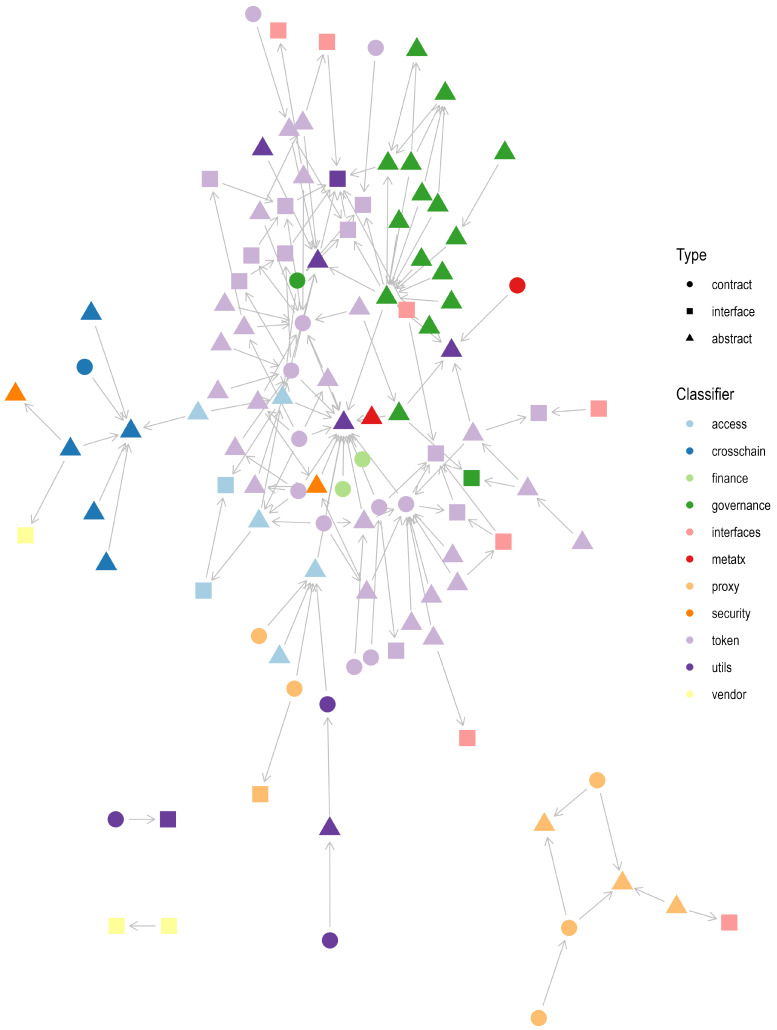
Inheritance of OpenZeppelin components containing contract types: (i) contracts, (ii) interfaces, and (iii) abstract contracts. Libraries have not been visualized. The most dominant classifiers are tokens, governance, and utilities.

**Figure 5 sensors-23-00639-f005:**
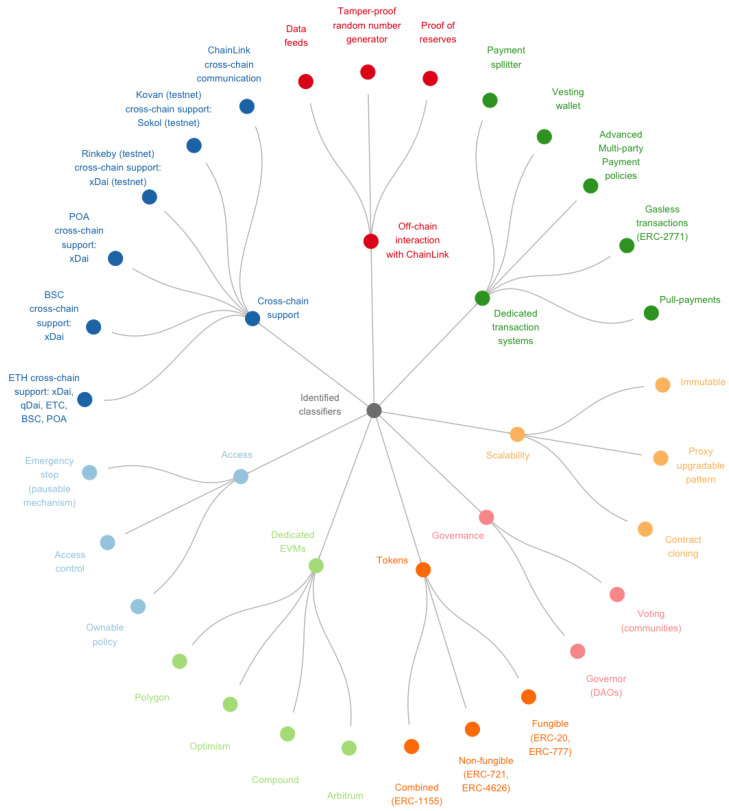
Identified classifiers represented into two levels: (i) main and (ii) detailed classifiers.

**Figure 6 sensors-23-00639-f006:**
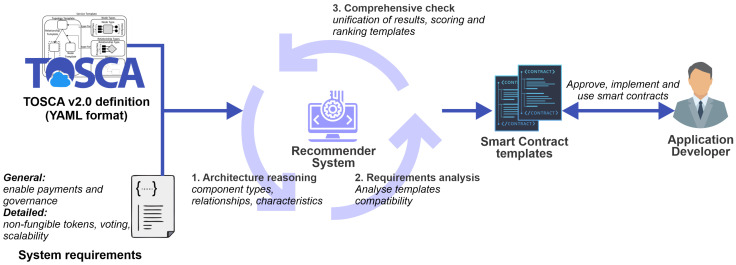
High-level architecture representing the pillar recommender system components with the main workflow.

**Figure 7 sensors-23-00639-f007:**
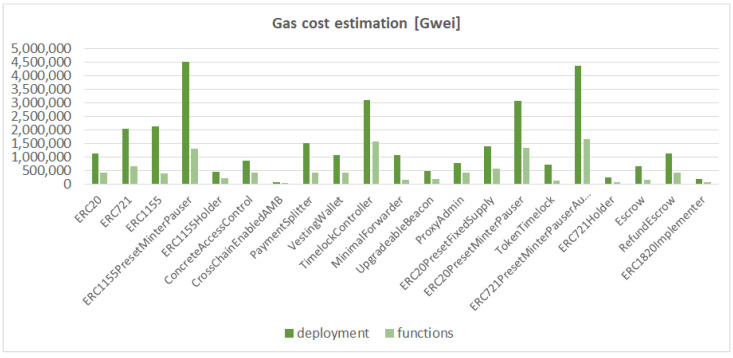
Cost estimation of the deployment process and available functions in deployable smart contracts.

**Table 1 sensors-23-00639-t001:** Comparison among the available EVM-compliant token standards.

Token Standard	Fungible Token	Non-fungible Token	Token Compatibility	Release Year	Advanced Features
ERC-20	Yes	No	/	2015	Reduces the complexity of token interactions.
ERC-721	No	Yes	/	2018	Enables a certificate of ownership for a virtual item.
ERC-777	Yes	No	extends ERC-20	2017	Allows backwards compatibility.
ERC-1155	Yes	Yes	functionalities from ERC-20, ERC-721 and ERC-777	2019	Allows batched operations for increased gas efficiency.
ERC-4626	Yes	No	extends ERC-20	2022	Standardizes tokenized vaults.

**Table 2 sensors-23-00639-t002:** Simulations of different recommender system scenarios.

IoT Fog Architecture	No. Components	Components Type	Requirements	Smart Contract Template Results
Azure IoT reference architecture	21	Application, Service, Gateway, Management, Device	Payment	ERC-20, ERC-777, ERC-4626, PaymentSplitter, VestingWallet
Intelligent Products Essentials reference architecture	19	Device, Application, Service, Gateway, Storage, Management	Fungible token, Governance	ERC-777, Governor, AccessControl
Smart Metering for Water Utilities	20	Device, Gateway, Management, Monitoring, Storage, Service, Hypervisor, Server	Token, Polygon, Scalability	ERC-4626, UpgradeableBeacon, IFxMessageProcessor

## Data Availability

Not applicable.

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
