# Peer review of "A Recommender System for Robust Smart Contract Template Classification"

_sensors, 2023, doi:10.3390/s23020639_

Round 1

Reviewer 1 Report

1. The abstract need to be rewritten not clear and the sequence of ideas are there.

2. In the introduction and a brief discussion of Smart contracts should be added. Moreover, the paragraphs are chaos and need to be rewritten.

3. The contribution is not clear.

4. An extensive English improvement is needed.

5. In the related work section “Smart contracts were first publicly available in 2015 with the release of the Ethereum decentralized ledger” redundant paragraph. the gap between the current contributions and the proposed approach of this manuscript should be added.

6. The complete workflow steps in section 3 are not clear.

7. smart contracts should be clarified as well as Ethereum decentralized ledger.

Author Response

We are thankful for these comments and suggestions because we believe that they have significantly contributed towards improving the manuscript.

Based on these comments and suggestions we performed the following revisions:

  • The abstract was rewritten to better highlight the goals and contributions. For that purpose, we followed the methodology of describing the paper context, objectives, method, and results.
  • The introduction section was rewritten in a more concise way. Moreover, we have included a brief discussion about smart contracts.
  • The contributions are highlighted in the abstract and in the last paragraph of the related work section.
  • The manuscript underwent meticulous correction of typos and grammar. The MDPI Language Editing Services were also used to improve the overall soundness of the paper.
  • The redundant content was removed and optimized throughout the paper. Moreover, the related work section was revised to be more cohesive.
  • The solution workflow was improved by updating Figure 1 to include more details and revising its description to contain more details and examples.  

All major revisions in the manuscript are marked in violet color for clarity.

Reviewer 2 Report

In this paper, to facilitate the development process of the smart contracts, the authors propose a recommender system that provides the smart contract developer with Ethereum Virtual Machine (EVM) based solidity based smart contract templates that match their requirements and are relevant for the typology of the Fog architecture. Experiment results demonstrate the effectiveness of the proposed scheme. However, I still have some concerns as follows.

1.What is the dataset used in the evaluation? More details could be provided.

2. What are the overheads (e.g., system setup, blockchain data storage and query, etc.) of the proposed algorithm?

3. There are more opportunities for conducting meaningful experiments to comprehensively evaluate the algorithm performance.

4. In the proposed recommender system, how to obtain a specific transaction record generated by smart contract? More explanation could be added.

5. More related technical papers about blockchain and IoT could be investigated. For example:

- Smartphone-assisted smooth live video broadcast on wearable cameras.

- BlockShare: A Blockchain empowered system for privacy-preserving verifiable data sharing.

- vChain+: Optimizing verifiable blockchain boolean range queries.

- VQL: Efficient and verifiable cloud query services for blockchain systems.

Author Response

We are thankful for these comments and suggestions because we believe that they have significantly contributed towards improving the manuscript.

Based on these comments and suggestions we performed the following revisions:

  • Information about the used datasets alongside references of the individual IoT architectures have been provided within in lines 422-424 of the manuscript:

“We performed simulations of our recommender algorithm on three base architectures (Azure 421 IoT reference architecture14, Intelligent Products Essentials reference architecture15 and Smart Metering for Water Utilities16) using different requirements, as shown in Table 2.”

  • Additional explanation related to relevant overheads such as time complexity related to the IoT fog architectures, used technologies, communication protocols and the database of the system have been explained in Section 6.2 (i.e. lines 405-410).
  • Additional experiments were performed in December 2022 to present the performance of the recommender system in the context of the smart contracts' development time, where 50 participants with different levels of blockchain expertise participated. The detailed description of the experiment evaluation is available in Section 6.3.
  • Additional explanation related to the records has been provided in the Section 5.2 where the description of the algorithm (lines 364-366) is aligned with the high-level architecture related step in the Section 6.2 (lines 410-420).
  • All proposed papers were properly included and referenced in our manuscript.

All major revisions in the manuscript are marked in violet color for clarity.

Round 2

Reviewer 1 Report

The list of contributions should be moved up to the introduction section before the paper structure section.

The authors claim that Algorithm 1 complexity is quadratic, they should discuss their claim.

Author Response

We are grateful for the reviewer's comments and suggestions. We have now updated the manuscript to address these minor but important suggestions. All changes in the text are in different color.

With kind regards,

Petar Kochovski

Reviewer 2 Report

All problems have been solved.

Author Response

We are grateful for the reviewer's aid with productive comments and suggestions. 

With kind regards,

Petar Kochovski